# Protecting the Human Rights of Refugees in Camps in Thailand: The Complementary Role of International Law on Indigenous Peoples

**Loi Thi Ngoc Nguyen**

Law School, University of Strathclyde, Glasgow G4 0LT, UK; nguyen-thi-ngoc-loi@strath.ac.uk

**Abstract:** This paper investigates whether and how International Law on Indigenous Peoples (ILIP) can complement protections granted under International Refugee Law (IRL) and International Human Rights Law (IHRL) to refugees in camps in Thailand. Presently, there are over 90,000 refugees from Myanmar in Thailand, confined to nine camps along the Thailand–Myanmar border. These refugees belong to different ethnic minority groups, but the vast majority are Karen—Indigenous Peoples from the Thailand–Myanmar border regions. They have fled to Thailand due to persecution by Myanmar authorities and segments of the Myanmar population. To date, Thailand has refused to become a party to the 1951 Refugee Convention or its 1967 Protocol. The country has failed to develop an asylum system and its laws continue to regard refugees as 'illegal migrants'. These refugees have been surviving in conditions of profound rightlessness. I posit that ILIP has a critical role to play in addressing the protection gaps and limitations in IRL and IHRL. In particular, the ILIP system of collective rights is vital in recognising the specific needs of refugees who are indigenous peoples. ILIP therefore provides a potent tool to make IRL and IHRL more responsive to the protection needs of indigenous refugees.

**Keywords:** refugees from Myanmar; refugee camps along Thailand–Myanmar border; Karen refugees; Thailand; refugee protection; human rights; International Law on Indigenous Peoples; collective rights; indigenous refugees; complementary role

## 1. Introduction

The provisions on international protection for refugees can be found in a range of legal sources and different fields of law with a diversity of rules. This diversity, which is a manifestation of the fragmentation of international law (Young 2015, p. 2), does not mean that norms from various areas of international law necessarily conflict. Rather, these norms overlap and can share a common goal and interact (ibid). In this regard, I argue that the interaction between International Law on Indigenous Peoples (ILIP), International Refugee Law (IRL) and International Human Rights Law (IHRL) forms a network of complementary protections for refugees.

IRL, which has two core instruments, the 1951 Refugee Convention and its 1967 Protocol,[1] constitutes the core of the international refugee protection regime. IRL, however, has proved of limited value to the many refugees who find themselves in camps in Thailand. Indeed, the country hosts 90,759 Myanmar[2] refugees in nine camps located in four provinces along the Thai–Myanmar border as of November 2022 (UNHCR Thailand

---

1  International Refugee Law (IRL) as used throughout my paper refers to:
   Convention relating to the Status of Refugees (adopted 28 July 1951, entered into force 22 April 1954) 189 UNTS 137 and Protocol relating to the Status of Refugees (adopted 31 January 1967, entered into force 4 October 1967) 606 UNTS 267.

2  Prior to 1989 the country was officially named Burma, after which the country's name officially changed to Myanmar. However, in the English-speaking world the country is often referred to as either Burma or Myanmar. This paper uses the terms Burma and Myanmar interchangeably throughout.

Multi-Country Office 2022), but still refuses to accede to the 1951 Refugee Convention or its 1967 Protocol.[3] Thus, when I use the term refugee(s) in relation to persons in camps in Thailand, I understand it in a wider sense than that arising from the 1951 Refugee Convention and its 1967 Protocol.[4] The term refugee brings to the fore these persons' need for international protection. When referring to those who have been granted refugee status within the meaning of the 1951 Refugee Convention and its 1967 Protocol, I employ the term recognised refugee(s). Because refugees in Thai camps have not been granted refugee status, they are denied the rights enshrined in the 1951 Refugee Convention (UN Human Rights Council 2021). Critically, in the absence of a national asylum system, Thai law views these refugees as 'illegal migrants'.

Although Thailand is not a state party to the 1951 Refugee Convention or its 1967 Protocol, the country is bound by IHRL instruments, which contain rights applicable to all human beings, including refugees in Thai camps. For this reason, I contend that the system of human rights conventions that Thailand is party to is instrumental in addressing IRL's limitations and gaps in protection. Refugees in Thai camps are therefore better protected under IHRL. However, below I show that IHRL is not best equipped to uphold the rights of Karen refugees in Thai camps as indigenous peoples. The vast majority of refugees presently in Thai camps are from the Karen peoples and are widely known as indigenous to the Thailand–Myanmar border region, although they are not legally recognised as indigenous peoples by Thailand and Myanmar (Lehman 1979; EthnoMed 2008). Yet, Karen refugees and the Karen people in general need their collective dignity and the value of their collective way of life to be recognised and protected (Howard 1995, pp. 83–84). It follows that ILIP has a critical role to play in protecting Karen refugees' collective rights as an indigenous people—something that IRL and IHRL cannot achieve on their own. ILIP promotes and preserves their unique collective cultures, values and traditions while seeking refuge in camps in Thailand.

In this article, I make the case for the complementary role of ILIP in the protection of indigenous refugees, and more specifically Karen refugees in Thai camps. I accept that there are still gaps—at times considerable—between internationally recognised rights and their enjoyment in practice. It is well established that the implementation and enforcement of international obligations continue to rest primarily with states, which makes implementation contingent on their 'goodwill' (Hannum et al. 2023; Bantekas and Oette 2020, pp. 25–26). However, failures in the implementation and enforcement of ILIP, IRL and IHRL do not negate the value inherent in investigating the role that the interaction between these three international legal regimes can play in buttressing protection for indigenous refugees. With this in mind, I analyse how this interaction can inform Thailand's legal and policy approach on refugee camps and thus advance protection for Karen refugees in camps along the Thai–Myanmar border.

I start by providing an overview of the historical background of the indigenous Karen and of their protracted refugee situation in Thai camps. I then discuss the respective role of each area of international law to apply, in particular how IHRL can fill the protection gaps arising from the failure of Thailand to sign up to the provisions of IRL, leaving refugees in Thai camps beyond its scope. As IHRL cannot fully protect the rights of Karen refugees as indigenous peoples, I then make the case for the key complementary role of ILIP in conferring protection on these refugees. This addresses a gap in the current literature in the

---

[3] For Thailand's ratification status to the 1951 Refugee Convention, see: https://treaties.un.org/Pages/ViewDetailsII.aspx?src=TREATY&mtdsg_no=V-2&chapter=5&Temp=mtdsg2&clang=_en (accessed on 5 December 2022).
For Thailand's ratification status to the 1967 Protocol Relating to the Status of Refugees, see: https://treaties.un.org/Pages/ViewDetails.aspx?src=TREATY&mtdsg_no=V-5&chapter=5&clang=_en (accessed on 5 December 2022).

[4] For the definition of refugee, see Article 1A (2) of the 1951 Refugee Convention. Firstly, the person must be outside his or her country of origin or habitual residence and have crossed an international border. Secondly, the person must have a well-founded fear of being persecuted for reasons of race, religion, nationality, membership of a particular social group, or political opinion. Thirdly, the person should be unable or unwilling to seek or take advantage of the protection of that country, or to return there.

field, where few studies consider how ILIP could bolster the rights of indigenous peoples in refugee situations. Indeed, while there is much literature on the protection of refugees, there is little that addresses indigenous peoples who are refugees.

## 2. Karen Refugees in Camps along the Thai–Myanmar Border: An Overview

### 2.1. The Karen as Indigenous Peoples: A Historical Background

While there are few written documents of the origins of the Karen peoples, Karen oral histories describe the arrival of their people as far back as 2500 years ago, after migrating through Tibet and China to present-day Myanmar (Minority Rights Group International 2017; McConnachie 2014, p. 24). By the eighteenth century, they were well established in the remote highland eastern region of Myanmar bordering Thailand and maintained autonomous village management systems (Minority Rights Group International 2017; Scott 2009; Renard 2003, p. 5).

The Karen form a population with various linguistic, sociocultural and religious backgrounds, with twelve sub-groups: Sgaw, Pwo, Pa-os, Paku, Maw Nay Pwa, Bwe, White Karens, Padaung (Kayan), Red Karen (Karenni), Keko/Keba, Black Karen and Striped Karen (Harriden 2002, p. 84; McConnachie 2014, p. 23). The vast majority of Karen are Buddhists (probably over two thirds), although large numbers converted to Christianity during British rule in Myanmar (Minority Rights Group International 2017). They have their own language, with the two most widely spoken Karen languages being Sgaw (predominantly spoken by Christian Karen) and Pwo (predominantly spoken by Buddhist Karen) (McConnachie 2014, p. 23).

It is well established that the Karen are indigenous to the Thailand-Myanmar border region. The Karen peoples, since their earliest history, have lived in autonomous villages in the eastern region of Myanmar bordering Thailand and have always considered themselves indigenous and different from the Burman group living in the lowlands of Myanmar— the dominant and largest ethnic group in Myanmar (Renard 2003, pp. 4–5; Mason 1862). The Karen, among other ethnic minority groups in Myanmar, are struggling to maintain and practice their own cultures including language and religion, as the central Burmese government aims to Burmanise them (Pedersen 2008, p. 56). The indigenous Karen peoples have been largely marginalised by the central Burmese state and many have engaged in a long, armed struggle for autonomy (ibid., pp. 47–48).

During the British colonial period in Myanmar, the Karen fought on the British side against the central Burmese state in order to secure their independence and autonomy (McConnachie 2014, p. 27; Taylor 2007, pp. 74–75; Ng 2022, pp. 189–90). With the British withdrawal in 1948, there were massive uprisings of the Karen against the central Burmese state (Lintner 1999, pp. 9–10; McConnachie 2014, p. 23). The Karen were the first ethnic group to take up arms against the central Burmese via the Karen National Union (KNU). The KNU is viewed as one of the oldest active insurgent groups in the world today, having fought the government for autonomy continuously since 1949 (Pedersen 2008, p. 48). Indeed, the self-determination movement of the Karen peoples is sometimes described as the world's longest running self-determination movement throughout history and is still in existence today (McConnachie 2014, p. 28).

### 2.2. The Protracted Refugee Situation in Camps in Thailand

Importantly, the current position of the Karen in protracted refugee situations in Thai camps has its origin in the long history of ethnic conflicts inside Myanmar and the fighting for self-determination as mentioned the section above (Clarke 2001, pp. 422–23). In particular, throughout the 1970s and early 1980s, the Burmese military followed a pattern of dry-season offensives and wet-season retreats, and ethnic minority villagers under attack echoed this movement, crossing into Thailand to escape a military offensive and returning when the troops departed (McConnachie 2014, p. 33). For the first time in 1984, Myanmar Army troops did not retreat when the rainy season came and large numbers of the Karen were trapped in Thailand, causing the creation of the first temporary refugee camps (ibid.). Since

then, as the Burmese government has not granted either political autonomy or substantial rights to ethnic minorities, the indigenous Karen have continued to flee to refugee camps in Thailand. The situation has been made worse in recent times when Myanmar's military, under the command of coup leader Gen. Min Aung Hlaing, seized power on the 1 February 2021 and launched a series of airstrikes in the areas of ethnic Karen people in Myanmar's southeastern region (Kapur 2022, p. 204; UN Office of the High Commissioner for Human Rights 2023; Gravers 2023; Refugees International 2021).

While the situation of political uncertainty in Myanmar remains dangerous, the country is not yet safe for refugees to return. Despite this, Thailand has, for decades now, continued to apply a hostile immigration policy to these refugees. Refugees are left in limbo, are not granted refugee status and are not allowed access to sufficient protection of basic human rights (UN Human Rights Council 2021). Refugees are confined in remote camps, are not able to leave the camps for work and are excluded from the Thai educational system (Human Rights Watch 2012, pp. 1–4; UN Human Rights Council 2021). Should refugees leave camps without official permission, they will be subjected to deportation (Human Rights Watch 2012, pp. 1–4). Refugees in camps cannot access the Thai healthcare system, and especially faced serious problems during the 2019 Coronavirus pandemic (COVID-19) including in relation to vaccines, tests, masks and disinfectants (UN Committee on the Elimination of Racial Discrimination 2022; Kobayashi et al. 2021).

The Karen, under Thai policy and law, have very limited control of their life choices in refugee camps. The Thai government restricts the teaching of history containing sensitive content, such as Karen revolutionary history or histories of Karen hardship, histories which are considered part of the indigenous Karen identity (Oh 2010, p. 7). Teaching materials containing critical historical and political education that might promote revolution or war in the refugee community in Thailand against their historic enemy, the Burmese government, are banned from use in camps (Oh 2012, p. 88). Karen refugee students in camps are not able to fully explore and understand their own history and their community. Young Karen refugees cannot gain knowledge and meaning from their indigenous collective heritage. The Karen refugees in camps are, in general, in a deeply vulnerable situation and are facing cultural erosion; their distinct collective culture and values are at risk of being diminished (Carpeño and Feldman 2015, pp. 417–18). I will now turn to explore the interaction of areas of international law in protecting refugees in Thai camps, and indeed will start from an analysis of the role of IHRL in filling the gaps and limitations of IRL in order to protect these refugees.

## 3. IHRL's Role in Addressing IRL's Limitations in the Protecting Refugees in Thai Camps

### 3.1. IRL and Its Limitations in the Protection of Refugees in Camps in Thailand

IRL developed in the post-Second World War period in order to support the displaced populations of Europe, and rests upon humanitarian premises (Barnett 2002, p. 246; Hathaway 2005, p. 91). Today, some three-quarters of the world's governments have bound themselves to respect the standards set by the 1951 Refugee Convention and its 1967 Protocol (Hathaway 2021, p. 171; Edwards 2018, pp. 539–40; UNHCR 2002).[5] Yet, whilst Southeast Asia is currently hosting a large population of refugees, the region has a very low level of ratification, with only two states of the Association of Southeast Asian Nations (ASEAN) having ratified the 1951 Refugee Convention and its 1967 Protocol, namely Cambodia and the Philippines (Moretti 2021, p. 214). As noted above, Thailand is not party to these IRL instruments.

---

[5] For State Parties including Reservations and Declarations to the 1951 Refugee Convention, see further: https://treaties.un.org/Pages/ViewDetailsII.aspx?src=TREATY&mtdsg_no=V-2&chapter=5&Temp=mtdsg2&clang=_en (accessed on 5 December 2022).
For State Parties Including Reservations and Declarations to the 1967 Protocol Relating to the Status of Refugees https://treaties.un.org/Pages/ViewDetails.aspx?src=TREATY&mtdsg_no=V-5&chapter=5&clang=_en (accessed on 5 December 2022).

Critically, the concept of a refugee does not exist in Thai law and the country has continually looked at issues pertaining to refugees as immigration law matters (Jetschke 2019, p. 712). For decades now, the primary objective of the Thai immigration system has been to deter migrants, including those who need international protection from entering and residing in Thailand (Gruß 2017, p. 25). Refugees in Thailand fall within the scope of the Thai Immigration Act B.E. 2522 (Immigration Act 1979) (Jetschke 2019, p. 712; Coddington 2018, p. 329).

It is important to note that in 2019, Thailand enacted the Regulation of the Office of the Prime Minister on the Screening of Aliens who enter into the Kingdom and are Unable to Return to the Country of Origin B.E. 2562 (the Regulation).[6] Clause 3 of this Regulation introduced a National Screening Mechanism (NSM), which would assess aliens who cannot return to their country of origin for 'Protected Person' classification. It is notable, however, that the Regulation does not grant refugee status within the meaning of the 1951 Refugee Convention and its 1967 Protocol (Chotinukul 2020, p. 27). Throughout the text of the Regulation, Thailand deliberately avoids using terms such as 'refugee' or 'asylum', and the legal status of the protected persons remains unclear (ibid.). In addition, although the Regulation came into effect in 2020, the onset of the COVID-19 pandemic slowed the implementation of the NSM (Stover 2021). The Regulation is therefore untested, and there are no reports of anyone having been granted protected person status (ibid.).

Refugees in Thailand still fall within the scope of the Immigration Act 1979 and are considered to be illegal migrants. Section 12(1) of Immigration Act 1979 accordingly provides that 'aliens' will be excluded from entering Thailand if they have no valid passport, travel document or visa stamped by a Thai authority. Those who enter Thailand without the requisite documentation are classified as illegal migrants. As those who seek refuge in Thailand often enter without papers and are unlikely to meet entry requirements, they are, under Section 12(1) of the Immigration Act 1979, categorised as illegal migrants (Jetschke 2019, p. 712). Indeed, refugees and asylum seekers in Thailand are considered to be the same as all other illegal migrants (Lego 2018, p. 184; Al Imran 2022, p. 985). As illegal migrants, refugees are, in accordance with Section 29 of Immigration Act 1979, sent out of Thai territories.

However, Section 17 of the Immigration Act 1979 also provides the Thai government with discretionary powers to allow people without the necessary documents to enter and stay in Thailand under some special circumstances. Interestingly, Section 17 of the Immigration Act 1979 does not specify which special circumstances may justify the exercise of the Thai government's discretionary power. On this legal basis, the Thai government exceptionally allows refugees from Myanmar, mostly indigenous groups fleeing from political persecution or fighting with the Burmese government, to enter the country as long as they stay within nine camps (officially designated 'temporary shelter') along the Thailand–Myanmar border (Vungsiriphisal et al. 2014, pp. 38–42; Petcharamesree 2016, p. 178).

To be clear, admission to Thai camps does not amount to being granted refugee status within the meaning of the 1951 Refugee Convention and its 1967 Protocol. This admission to Thai camps is not granted with a view to resettlement in Thailand (Human Rights Watch 2012, p. 18; UNHCR 2006). Instead, the Thai government views the stay of refugees in Thai camps as only a temporary matter and assumes that they should prepare for resettlement in third countries or repatriation to Myanmar (Brees 2008, p. 384). These people in camps under Thai Immigration law retain the status of illegal migrant (Human Rights Watch 2012, pp. 18–19; UNHCR 2006).

Critically, as refugees in Thai camps have not been granted refugee status and are, under Thai law, considered illegal migrants, they are not able to access the range of refugee rights articulated in the 1951 Refugee Convention (UN Human Rights Council 2021). In particular, the 1951 Refugee Convention and its 1967 Protocol define the term refugee and set out a range of basic rights attached to the status of refugees, such as the right to

---

6    Regulation of the Office of the Prime Minister on the Screening of Aliens who Enter into the Kingdom and are Unable to Return to the Country of Origin (entered into force 24 December 2019) B.E. 256225.

work, the right to education, the right to free movement within the state that has bestowed refugee status, and other rights (Goodwin-Gill 2016, pp. 36–37).

The range of rights enshrined in the 1951 Refugee Convention consequently only apply to persons who have been granted refugee status (Chetail 2021, p. 208). Indeed, these rights are not applicable either to asylum seekers or to peoples who are at risk in their own countries but are not recognised as refugees within the definition of the Refugee Convention (ibid.). It is clear here that because people from Myanmar in camps under Thai Immigration law are, as analysed above, considered to be illegal migrants, they cannot enjoy the range of rights in the 1951 Refugee Convention and its 1967 Protocol, which are conferred only upon recognised refugees (Human Rights Watch 2012, pp. 18–19; UNHCR 2006).

The only provision of the 1951 Refugee Convention that applies to both recognised refugees and all asylum seekers, including refused asylum seekers, is the principle of *non-refoulement* articulated in Article 33(1) (UNHCR Executive Committee 1977). Accordingly, state parties are not allowed to return a refugee to a country where their life or freedom would be threatened on the basis of their race, religion, nationality, membership of a particular social group or political opinion.

It is important to note that the prohibition against the *refoulement* of refugees stipulated in Article 33(1) of the 1951 Refugee Convention has acquired the status of a norm of customary international law (UNHCR 1997, 2002). The fundamentally norm-creating character of the principle *non-refoulement* (*opinio juris*) is supported by the fact that the principle receives extensive citation in many Conclusions of the Executive Committee of UNHCR and in a number of important binding and nonbinding international texts (Lambert 2021, p. 245; Lauterpacht and Bethlehem 2003, pp. 143–44).[7] This cornerstone of IRL is therefore legally binding upon Thailand and all other States which have not ratified the 1951 Refugee Convention and/or its 1967 Protocol (UNHCR 2001, p. 14; Lambert 2021, p. 240).

However, the principle of *non-refoulement* articulated in Article 33(1) of the 1951 Refugee Convention is not absolute and has exceptions (Duffy 2008, p. 374). In accordance with Article 33(2) of the 1951 Refugee Convention, the benefit of the present provision may not be claimed by asylum seekers or refugees whom there are reasonable grounds to regard as a danger to the security of the asylum country, or who constitute a danger to the community of that country. Overriding reasons of national security or public safety will allow states to derogate from this principle and permit lawful refoulements (Lauterpacht and Bethlehem 2003, p. 155).

It is important to emphasise that although Thailand is bound by the principle of *non-refoulement*, the absence of formal asylum procedures and, more generally, the lack of refugee law and policy, produce an environment that does not account for the obligation to comply with the principle of *non-refoulement*. Since they are illegal migrants, should refugees leave the camps without official permission, they can be subjected to deportation (Human Rights Watch 2012, pp. 1–4). This clearly violates the principle of *non-refoulement*—the cornerstone of IRL.

---

7  For example, the principle of *non-refoulement* is cited in the following documents: UNHCR Executive Committee. 1996. General Conclusion on International Protection No. 79 (XLVII). A/51/12/Add.1. Available online: https://www.refworld.org/docid/3ae68c430.html (accessed on 15 November 2022); UNHCR Executive Committee. 1997. General Conclusion on International Protection No. 81 (XLVIII). A/52/12/Add.1. Available online: https://www.refworld.org/docid/3ae68c690.html (accessed on 15 November 2022).
Declaration on Territorial Asylum, UNGA res 2312 (XXII) (adopted 14 December 1967), Article 3; Organisation of African Unity Convention Governing the Specific Aspects of Refugees Problems in Africa (adopted 10 September 1969, entered into force 20 June 1974) 1001 UNTS 45, Article 2; Final Text of the Asian-African Legal Consultative Organization (AALCO)'s 1966 Bangkok Principles on Status and Treatment of Refugees (adopted on 24 June 2001), Article 3(1).
Also including the various expressions by the Council of Europe such as:
Council of Europe. 1967. Committee of Ministers, Resolution (67) 14: Asylum to Persons in Danger of Persecution. Available online: https://www.refworld.org/docid/3ae6b38168.html (accessed on 10 September 2022); Council of Europe. 1984. Committee of Ministers, Recommendation No R (84) 1 on the Protection of Persons Satisfying the Criteria in the Geneva Convention Who Are Not Formally Recognised as Refugees. Available online: https://www.refworld.org/docid/3ae6b3816c.html (accessed on 11 November 2022).

It is undeniable that IRL is the central international legal regime in protecting refugees, but this remains limited in the context of refugees in camps in Thailand given that the Thai government has refused to ratify the 1951 Refugee Convention and its 1967 Protocol. In addition, although Thailand is not party to the 1951 Refugee Convention and its 1967 Protocol, the country as a member State of the United Nations is obligated to cooperate with the UNHCR in the fulfilment of its responsibilities to protect refugees in camps.[8] However, the Thai government has not been willing to cooperate with the UNHCR and has continually reduced the role of UNHCR, particularly in the Thai–Burma border refugee camps (McConnachie 2012, p. 40). The UNHCR plays a minimal role in supporting and protecting of refugees in Thai camps (ibid.).[9] The next section will now turn to analyse how IHRL is instrumental in addressing the limitations of the applicability of IRL in Thailand in the protection of their rights as refugees.

### 3.2. IHRL's Role in Complementing Protection under IRL for Refugees in Camps in Thailand

While Thailand remains a non-signatory of the 1951 Refugee Convention and its 1967 Protocol, the country importantly is party to core human rights treaties, such as the International Covenant on Civil and Political Rights (ICCPR), International Covenant on Economic, Social and Cultural Rights (ICESCR), the 1984 Convention against Torture and other Cruel, Inhuman or Degrading Treatment or Punishment (CAT) and the Convention on the Rights of the Child (CRC).[10] With this in mind, I examine and evaluate IHRL's contributions to protecting refugees in Thai camps.

Firstly, while most of the rights contained in the 1951 Refugee Convention and its 1967 Protocol are, as mentioned in Section 3.1, only granted to recognised refugees, the rights enshrined in IHRL are plainly applicable to all persons, regardless of their immigration or other status (Edwards 2018, pp. 539–40; Harvey 2015, pp. 43–44). In particular, Article 2 of the Universal Declaration of Human Rights (UDHR)[11] emphasises the principle of non-discrimination, specifying that every human being has inherent dignity and is entitled to all the rights and freedoms set forth in the Declaration. Human rights should be given to everyone without distinctions of any kind, such as race, colour, sex, language, religion, political or other opinion, national or social origin, property, birth or other status.[12] The principle of non-discrimination has become a core principle and is frequently cited in the range of subsequent human rights treaties, including Article 2(2) of the ICESCR and Article 2(1) of the ICCPR or Article 2(1) of the CRC.

Although they have not been bestowed refugee status, refugees in Thai camps are bestowed rights under IHRL as human beings, irrespective of their immigration status. For instance, the right to employment is, in accordance with Article 17(1) of the 1951 Refugee Convention, limited only to recognised refugees. In contrast, Article 6 of the ICESCR that Thailand is party to provides that everyone is entitled to freely choose their work and obliges state parties to take appropriate steps to safeguard this right. The UN Committee on Economic, Social and Cultural Rights (CESCR) has further emphasised that the right

---

8　For more information on the obligation of states of the United Nations to cooperate with the UNHCR, see: Statute of the Office of the United Nations High Commissioner for Refugees, UNGA res 428 (V) (adopted 14 December 1950).

9　Within the limited scope of this research, the paper will not discuss the governance architecture of the Thai–Burma border refugee camps, including the role of UNHCR, in depth.

10　International Covenant on Civil and Political Rights (adopted 16 December 1996, entered into force 23 March 1976) 999 UNTS 171; International Covenant on Economic, Social and Cultural Rights (adopted 16 December 1966, entered into force 3 January 1976) 993 UNTS 3; Convention Against Torture and Other Cruel, Inhuman or Degrading Treatment or Punishment (adopted 10 December 1984, entered into force 26 June 1987) 1465 UNTS 85; Convention on the Rights of the Child (adopted 20 November 1989, entered into force 2 September 1990) 1577 UNTS 3.
For more information on the ratification status for Thailand, see further at UN Treaty Body Database. Available online: https://tbinternet.ohchr.org/_layouts/15/TreatyBodyExternal/Treaty.aspx?CountryID=172&Lang=EN (accessed on 28 November 2022).

11　Universal Declaration of Human Rights, UNGA res 217A (III) (adopted 10 December 1948).

12　Article 2 of the UDHR.

to work articulated in Article 6 of the ICESCR applies to everyone including refugees, asylum seekers, stateless persons, migrant workers and victims of international trafficking, regardless of legal status and documentation (UN Committee on Economic, Social and Cultural Rights 2009, p. 9, para. 30).

The CESCR also explicitly acknowledges the vulnerability of refugees due to their often-precarious legal status and accordingly asserts that Contracting States should enact legislation enabling refugees to work and under conditions no less favourable than nationals (UN Committee on Economic, Social and Cultural Rights 2016, p. 13, para. 47(i)). Here, it is clear that although refugees in Thai camps have not been granted refugee status and are, under the 1951 Refugee Convention, not allowed to work, they as human beings are, in accordance with Article 6 of the ICESCR, entitled to this fundamental right. Thailand is therefore obliged, under IHRL, to grant them the right to work and crucially, take appropriate measures to ensure that refugees in camps are able to exercise or engage in employment in practice.

Moreover, Thailand in accordance with Article 13(2) of ICESCR, has an obligation to make primary education compulsory and available free to all persons including refugees in camps, without reference to nationality or immigration status. Thailand must make secondary education in its different forms including technical, vocational training available and equally accessible to all by every appropriate means.[13] Higher education also should be made equally accessible to all. Indeed, refugees in camps are under IHRL, entitled to equal treatment with Thai nationals with respect to free and compulsory primary education and to access different forms of secondary and higher education. Thailand is not allowed to exclude refugees in camps from the Thai school system. Furthermore, Thailand is under Article 12(2) of ICESCR, required to adopt and implement measures ensuring the right of access to health facilities, goods and services to all on a non-discriminatory basis. Refugees in camps should accordingly be eligible to access all Thai medical services including prevention and treatment, including for diseases such as COVID-19.

Another way in which IHRL is instrumental in complementing IRL in the protection of refugees in camps in Thailand can be seen through the application of the principle of *non-refoulement*. In addition to the 1951 Refugee Convention and its 1967 Protocol, the principle of *non-refoulement* is also expressed in international human rights treaties. Although the human rights principle of *non-refoulement* largely coincides in substance with the refugee law principle of *non-refoulement*, the former offers broader protection than the latter, and has no exemptions (Chetail 2021, p. 209). Indeed, Article 33(1) of the 1951 Refugee Convention states that refugees are protected against return to a country where their life or freedom would be threatened on account of their race, religion, nationality, membership of a particular social group or political opinion. As analysed in Section 3.1, this principle articulated in Article 33(1) of the 1951 Refugee Convention is not absolute. Article 33(2) of the 1951 Refugee Convention also permits lawful refoulement in specific circumstances such as the existence of a danger to the security of the country or a danger to the community of the country.

This is obviously in contrast with the human rights principle of *non-refoulement*. In particular, Article 3 of the CAT states that countries shall not expel or return a person to another state where there are substantial grounds for believing that doing so would expose them to a danger of being subjected to torture. Article 7 of the ICCPR also mentions that no one shall be subjected to torture or to cruel, inhuman or degrading treatment or punishment. Although Article 7 of the ICCPR does not explicitly forbid refoulement to such ill treatment, the Committee on Civil and Political Rights (CCPR) has interpreted this provision as concluding a prohibition on removal of peoples to places of torture, cruel, inhuman, humiliating or degrading treatment or punishment (UN Human Rights Committee 1992, p. 2, para. 9; UN Human Rights Committee 2004, p. 5, para. 12).

---

13    Article 13(2) of ICESCR.

Indeed, the principle of *non-refoulement* articulated in the human rights treaties is present to protect all peoples irrespective of their immigration status from suffering severe forms of ill-treatment that cause serious harms to all human life. Importantly, the prohibition on refoulement under IHRL is absolute and has no derogation (UN Human Rights Committee 1992, p. 1, para. 3; UN Committee Against Torture 2008, p. 2, para. 5–6). This means that even if a refugee or asylum seeker could be returned in accordance with Article 33(2) of the 1951 Refugee Convention, IHRL may still not permit refoulement on the humanitarian grounds of the prohibition of torture, cruel, inhuman, humiliating or degrading treatment or punishment and other irreparable harm (Edwards 2018, p. 549; Goodwin-Gill and McAdam 2007, p. 306; Mathew 2021, p. 903; International Committee of the Red Cross 2017, p. 348).

Therefore, IHRL offers refugees from Myanmar in Thai camps stronger protection against *refoulement* than IRL. Thailand, as a state party to both the ICCPR and CAT, is under no circumstances allowed to view refugees in camps as subjects for deportation and is not permitted to forcibly return refugees to Myanmar where they may face the danger of being subjected to torture or ill treatment. Any deportation of refugees to Myanmar due to leaving camps without official permission given by the Thai authorities is illegal under IHRL.

Although Thailand is a signatory to international human rights instruments and is bound by their provisions, the country's implementation of the human rights obligations in practice as mentioned at the start of this paper continues to pose a challenge. In particular in accordance with the 2017 Constitution of Thailand,[14] the country applies a dualist approach to the incorporation of treaties into its domestic legal system. Under a dualist system, international law and national law are considered separate legal systems wherein the rules and obligations of international law binding upon the state do not automatically become a part of national law (Verdier and Versteeg 2015, p. 516).

Section 178 of the 2017 Constitution of Thailand specifies that there is no treaty that has direct applicability in Thailand. For treaties to become law in the municipal sphere, it requires the enactment of an Act for implementation approved by the National Assembly. Consequently, the international human rights treaties that Thailand is party to are only implemented if the Thai government has transformed or incorporated them into domestic law. The implementation of international human rights obligations remains therefore entirely at the goodwill of the Thai government. This has led to concern that refugees in Thai camps might not actually benefit from provisions under IHRL, even including ones to which Thailand has signed up.

However, it is important to understand that, in the decades since the end of World War II, a normatively robust human rights regime has been developed and shaped by an ideal that human beings are born free and equal in dignity and rights (Morsink 2019). IHRL has been widely endorsed because its normative force is inescapable in the contemporary world (Donnelly and Whelan 2020). All states have accepted that human rights are a legitimate subject of international politics (ibid). I argue that even though the likelihood of the Thai government implementing its obligations arising from human rights instruments may be low, the provisions of IHRL remain fundamental to the protection of refugees in Thai camps and constitute one of the core approaches to inform the critique and development of law and policy towards these refugees. The reality also remains that, even in Thailand's dualist system, once the international human rights treaties are signed, Thailand is subject to obligations that bind them on the international plane. Signing up to human rights instruments, being bound by their provisions, is the first important step in their later implementation in domestic law.

It is clear that human rights treaties provide a unique and vital source of refugee protection in the 43 United Nations Member States, including Thailand, that have not ratified the 1951 Refugee Convention and its 1967 Protocol (Chetail 2021, p. 203). IRL and IHRL work hand in hand, and complement and reinforce each other within one single

---

[14] The Constitution of the Kingdom of Thailand (entered into force on 6 April 2017) B.E. 2560.

continuum of protection. Provisions of both of these legal regimes working together will offer strong standards of treatment that the refugee policies of many states today should comply with and in particular, provide effective protections for rights for refugees in camps in Thailand. However, I argue that, as Karen refugees in Thai camps are indigenous peoples, provisions of IRL and IHRL do not provide adequate protection of their indigenous rights and needs. In the next section below, I will now turn to analyse the role of ILIP and how ILIP complements provisions of IRL and IHRL in the protection of indigenous refugees in camps in Thailand.

## 4. ILIP's Role in Complementing Protections under IRL and IHRL for Refugees in Thai Camps

While fleeing from their homeland and living in the refugee camps in Thailand, the indigenous Karen peoples continue to seek to protect their own community and enhance their autonomy as well as the integrity of their own distinct indigenous identity (McConnachie 2014, pp. 46–51).[15] They often claim collective rights which are indispensable for their existence, well-being and integral development as indigenous groups.

Critically, the provisions of IRL and IHRL as analysed in previous sections mainly focus on individual rights rather than group rights and necessarily protect the Karen refugees as individuals rather than groups. Although the UNHCR accepts refugees on a prima facie basis, for example in large-scale refugee situations, members of that group are considered individually as refugees and the system of rights that attach to them under the 1951 Refugee Convention and its 1967 Protocol remain individual rights, held individually by the members of that group (UNHCR 2015). That said, IHRL offers some level of protection for group rights; for example, Article 27 of the ICCPR recognises the right of groups to enjoy their communal culture, profess their religion and use their language.

It cannot be denied that IRL and IHRL alone cannot, however, fully protect the specific needs of the Karen refugees. This is clearly the case, especially when these indigenous refugees are, as mentioned in Section 2, left in a deeply vulnerable situation in a protection vacuum and exposed to the risk of cultural erosion and identity loss. Even when refugee status is granted to these Karen refugees, the international standards on the subject, particularly IRL and IHRL do not provide the necessary specific protection that guarantees the preservation of the cultural identities of the indigenous refugees (Figueira 2020, p. 443). In the face of profound vulnerability, Karen refugees are in need of the indigenous collective rights framework articulated in ILIP in addition to and beyond the system of rights of IRL and IHRL.

To date, the 1989 Convention on Indigenous and Tribal Peoples (ILO Convention 169)[16] and the United Nations Declaration on the Rights of Indigenous Peoples (UNDRIP)[17] are the main international instruments on the protection of indigenous peoples (Lennox and Short 2016, p. 5).[18] ILO Convention 169 is the only legally binding international treaty on indigenous peoples (ibid.). It has been ratified by 24 countries, which do not include Thailand.[19] It is important to emphasise that, although reluctance by States including Thailand to ratify is indicative of the existing challenge of ILO Convention 169, it is a fact that ILO Convention 169 has led to profound changes in the domestic legal systems of ratifying countries (Ormaza and Oelz 2020, p. 73). It remains the only treaty open

---

[15]  It is noted that despite the limited opportunities available in refugee camps, and the restrictions of the Thai government, the indigenous Karen still attempt to build dynamic Karen communities and structure their daily life in camps in the way of their traditional village and community life (McConnachie 2014, p. 45).

[16]  C169 Indigenous and Tribal Peoples Convention (adopted 27 June 1989, entered into force 5 September 1991) 1650 UNTS 383.

[17]  Declaration on the Rights of Indigenous Peoples UNGA res 61/295A (adopted 13 September 2007).

[18]  I acknowledge that there are also other related Conventions such as The Convention on Biological Diversity (CBD) or The United Nations Framework Convention on Climate Change (UNFCCC). However, within the limited scope of this research, I cannot discuss all, instead focusing on the core documents directly relevant to the situation of Karen indigenous refugees. To be clear, International Law on Indigenous Peoples (ILIP) as used throughout my paper refers to the ILO Convention 169 and UNDRIP.

[19]  For more information on ratifications of the ILO Convention 169, see at: https://www.ilo.org/dyn/normlex/en/f?p=NORMLEXPUB:11300:0::NO::P11300_INSTRUMENT_ID:312314 (accessed by 15 November 2022).

for ratification specifically and exclusively dedicated to the promotion and protection of indigenous peoples' rights and culture (Ormaza and Oelz 2020, p. 72; Swepston 2018, p. 3). The importance and contribution of ILO Convention 169 have become prominent.

ILO Convention 169 was the result of the revision of the preceding ILO Convention on Indigenous and Tribal Populations (ILO Convention 107)[20] (Thornberry 2002, p. 27; Wolfrum 1999, pp. 371–72). ILO Convention 169 importantly lays down comprehensive protection standards for indigenous peoples; it explicitly aims at removing the assimilationist orientation of the earlier standards in ILO Convention 107.[21] ILO Convention 169 instead emphasises the aspirations of indigenous peoples to exercise control over their own institutions, education, ways of life and economic development and to maintain and develop their own identities, languages and religions.[22] ILO Convention 169 calls on states to value the distinctive contributions of indigenous peoples to the cultural diversity of humankind.[23]

Within this framework, ILO Convention 169 recognises indigenous peoples as 'peoples' and takes a decisive stand on the collective nature of indigenous rights by emphasising a series of provisions on collective rights (Rodriguez-Pinero 2005, p. 321). Accordingly, indigenous peoples have rights to maintain and develop their own societies. States are urged to respect, recognise and protect the social, economic and cultural identities, and the customs and traditions and institutions of indigenous peoples[24] as well as to respect the integrity of these values, practices and institutions.[25] More specifically, ILO Convention 169 recognises rural and community-based industries, as well as subsistence economies and traditional activities of indigenous peoples such as hunting, fishing, trapping and gathering as important factors in the maintenance of their cultures and in their economic self-reliance and development.[26]

The cornerstone of ILO Convention 169 rests in the participatory rights of indigenous peoples (Yupsanis 2010, p. 438). Article 6(1)A of ILO Convention 169 provides that states shall consult indigenous peoples through appropriate procedures, whenever consideration is being given to legislative or administrative measures which may affect them directly. The consultation must be undertaken in good faith.[27] Article 7(1) of ILO Convention 169 further strengthens the possibility of indigenous peoples' participation in decisions that concern them, such as their right to decide their own priorities for the process of development affecting their own lives, beliefs and institutions or their right to exercise control over their own economic, social and cultural development.

While ILO Convention 169 is a legally binding international treaty, UNDRIP is a non-binding instrument (usually known as a soft law) (Lennox and Short 2016, p. 5). Although UNDRIP is a non-binding instrument, it represents a global consensus on the standards relating to indigenous peoples and is considered as a key international legal document on the rights of indigenous peoples (Odello 2016, p. 64). Interestingly, UNDRIP is the product of indigenous peoples and their insistence on the inclusion of articles that responded to their needs (Burger 2016, p. 322). It is one of the very few UN legal documents that have been elaborated in consultation with the victims of human rights abuses and with peoples who are to be the beneficiaries (ibid.). Despite its non-binding nature, provisions of UNDRIP therefore play a key role in shaping policy and law towards indigenous peoples around the world, including the case of indigenous Karen refugees in Thai camps.

It is also important to note here that soft law and binding instruments such as treaties or customary law can interact and build upon each other as complementary tools for solving

---

20 C107 Indigenous and Tribal Populations Convention (adopted 26 June 1957, entered into force 2 June 1959) 328 UNTS 247.
21 The preamble of the ILO Convention 169, paragraph 4.
22 The preamble of the ILO Convention 169, paragraph 5. See further: Article 5(A) and (C) of the ILO Convention 169.
23 The preamble of the ILO Convention 169, paragraph 7.
24 Article 2(2)(B) and Article 5(A) of the ILO Convention 169.
25 Article 5(B) of the ILO Convention 169.
26 Article 23(1) of the ILO Convention 169.
27 Article 6(2) of the ILO Convention 169.

international problems (Shaffer and Pollack 2010, p. 721). Soft law instruments are not law per se and thus have less legal effect than legally binding instruments (Focarelli 2019, p. 223). However, soft law instruments may acquire binding legal character as elements of a treaty-based regulatory regime or constitute part of a subsequent agreement between parties regarding the interpretation of a treaty or the application of its provision (Boyle 2014, p. 119).[28] Some soft-law instruments are also important as they can become a first step in a process eventually leading to the conclusion of a multilateral or regional treaty (Shaw 2021, p. 100; Boyle 2014, p. 123). Non-binding instruments can, with evidence of opinio juris (the belief that action is legally necessary), and widespread practice amongst states, facilitate the progressive evolution of customary international law (Boyle 2014, pp. 130–33).

The importance of UNDRIP has especially become clear as some provisions of UNDRIP may acquire the status of customary international law binding all states including Thailand (Wiessner 2012, pp. 54–56; Odello 2016, p. 64). In particular, although UNDRIP is a non-binding instrument, it was supported by an overwhelming majority of states, with 143 states including Thailand in favour[29] and since adoption in 2007, there is significant emerging practice relating to UNDRIP (Isa 2019, p. 15). Moreover, UNDRIP has been referred to by the Inter-American Court of Human Rights[30] and the African Commission on Human and Peoples' Rights and its Court.[31] They both repeatedly cite the provisions of UNDRIP and use UNDRIP as the legal basis for their findings and decisions (Isa 2019, p. 14; MacKay 2018). Domestic courts have also made use of UNDRIP (ibid., p. 15). For example, the Constitutional Court of Peru[32] and the Supreme Court of Belize[33] have used UNDRIP in some of their decisions. The Supreme Court of Belize indeed emphasised that Belize voted in favour of the Declaration and is not expected to disregard it. The Declaration has also been used to develop specific national laws and amend existing legislation in some countries (Odello 2016, p. 64; Isa 2019, p. 15). Most significantly, Bolivia explicitly incorporated UNDRIP into Bolivia's National Law 3897 of 26 June 2008 and recognised indigenous peoples' rights (Odello 2016, p. 64). Ecuador is another leading example that used the indigenous language of UNDRIP in the Constitution of 2008.[34] These examples have indeed shown an evolution of international consensus towards acknowledging the rights of indigenous peoples as set out in UNDRIP (Odello 2016, p. 64).

Although most rights embodied in UNDRIP are in general not new and are already part of the existing set of fundamental human rights included in other legally binding instruments such as ILO Convention 169 (Odello 2016, p. 64), there are also major innovations in UNDRIP (Isa 2019, p. 10). One of the major innovations put forward by UNDRIP is the recognition of the right to self-determination—one of the key demands by the global indigenous movement (ibid.). Article 3 of UNDRIP accordingly states that indigenous peoples have the right to self-determination. By virtue of that right, indigenous peoples freely determine their political status and freely pursue their economic, social and cultural development. Article 4 of UNDRIP continues to explain further that the exercise of the right to self-determination of indigenous peoples only takes place through autonomy and self-government in matters relating to their internal and local affairs.

---

28    See also: Vienna Convention on the Law of Treaties, (adopted 22 May 1969, entered into force 27 January 1980) 1155 UNTS 331, Article 31(3)(A).

29    See further the voting record for UNDRIP including the view of Thailand at: https://press.un.org/en/2007/ga10612.doc.htm (accessed on 20 December 2022).

30    For example: In *the Saramaka People vs. Suriname*, Preliminary Objections, Merits, Reparations and Costs, Judgment of 28 November 2007, Series C No.172 or in another recent case *Kaliña and Lokono Peoples vs. Suriname*, IACTHR, 2015, Series C, No. 309.

31    For example: In *African Commission on Human and Peoples' Rights vs Republic of Kenya*, Application No.006/2012 Judgment of 26 May 2017, paragraph 209.

32    For example: In *Tres Islas indigenous community Case*, Sentencia del Tribunal Constitucional, Exp. No. 01126-2011-HC/TC, Judgement of 11 September 2012.

33    For example: In *Aurelio Cal et al vs. Attorney General of Belize*, (Claim No. 17 and 172 of 2007), Judgement of 18 October 2007 (Mayan land rights).

34    See further at: Ecuador, Constitution, *Registro Oficial* 449, 20 October 2008, Articles 16, 29, 347, 379.

I acknowledge that whether or not provisions of UNDRIP have achieved customary law status remains an open question. However, the provisions of UNDRIP in particular may come to have a large role in the shaping of an international consensus and the future development of customary law. In addition, although Thailand is not a signatory to ILO Convention 169, its provisions remain important as a key benchmark for the treatment of indigenous peoples including the Karen refugees in Thai camps. I argue that Karen refugees as indigenous peoples should be eligible for the specific system of indigenous collective rights under ILO Convention 169 and UNDRIP. In accordance with this, Karen peoples in refugee camps should be allowed to practice, develop and teach their indigenous religious traditions, customs, ceremonies and history. They should have the right to establish and control their own educational systems, institutions and facilities and to ensure teaching in their Karen languages, in respect of their own collective cultures. They should also have the right to maintain and develop their own political, economic and social systems and be free to express their political status. Young Karen refugees should be allowed to learn knowledge and meaning from their indigenous collective heritage and to continue to preserve their own distinct values and tradition while staying in camps.

By bringing ILIP into the mix, I also understand that there is much debate that the ILIP system of indigenous collective rights are in opposition to the individual rights contained within IRL and IHRL (Patton 2016). The entitlement to indigenous collective rights may undermine their enjoyment of the system of individual rights articulated in IRL and IHRL (Ivison et al. 2000, pp. 1–5). However, I contend that these two systems of rights should not be seen as conflicting. I argue that the indigenous collective rights articulated in ILIP are of such a nature that indigenous peoples can choose the extent of their participation in them.

The indigenous collective rights articulated in ILIP only seek to enhance their group life and experience, but still preserve the right of indigenous peoples to deviate or exit from that group life should they so choose. This is indeed explicitly stated in the preamble of UNDRIP, emphasising that indigenous individuals are entitled without discrimination to all the human rights recognised in international law, and that these indigenous peoples at the same time possess collective rights which are indispensable for their existence, well-being and integral development as peoples.

Indigenous collective rights under ILIP are given to the Karen refugees on the basis of preserving their own cultures, values and traditions while seeking refuge in camps in Thailand. These indigenous group rights however should not be understood as being in opposition to the individual rights contained within IRL or IHRL. Granting these indigenous group rights would not prevent the Karen refugees from the enjoyment of protection under IRL and IHRL. Karen refugees have the right to access to the Thai education and healthcare systems or have the right to engage in Thai labour market. ILIP constitutes another layer of protection that complements IRL and IHRL and would create a protection regime more responsive to the rights and needs of refugees in camps in Thailand, especially for the special needs of Karen indigenous refugees.

## 5. Conclusions

In this paper, I have demonstrated that ILIP has a critical role to play in complementing protections IRL and IHRL for refugees in camps in Thailand. I have stressed that the primary consequence of Thailand's failure to ratify the 1951 Refugee Convention and its 1967 Protocol is that IRL remains of limited value in protecting refugees in Thai camps. Indeed, these refugees have not been granted refugee status and therefore are not bestowed the rights attached to this status under IRL. I then argued that although Thailand is not party to the 1951 Refugee Convention and its 1967 Protocol, it is bound by obligations enshrined in the international human rights instruments it has ratified. Importantly, the range of rights granted by IHRL are conferred on all human beings regardless of immigration status, which includes refugees in Thai camps. It follows that IHRL contributes to the protection of refugees in Thai camps as it addresses some of the limitations of IRL. However, I have also shown that IHRL is not best placed to uphold the rights of Karen refugees as an

indigenous people. The shortcomings of IRL in this respect are even greater. On this basis, I have argued that ILIP has a vital role to play in filling this protection gap as it recognises Karen refugees' group rights as members of an indigenous people. I have shown that the range of collective indigenous rights enshrined in ILIP do not conflict with the system of individual rights under IRL and IHRL. Rather, ILIP interacts and complements IRL and IHRL in protecting and promoting the distinct values and identities of Karen indigenous groups while staying in refugee camps in Thailand. By bringing ILIP into dialogue with IRL and IHRL, this paper brings to the fore the specific protection needs of Karen refugees as members of an indigenous people and how these can be best addressed. Importantly, while the paper has focused on Karen refugees, the proposed approach with its emphasis on the complementary role of ILIP is of relevance to other indigenous peoples who find themselves in a similar predicament to Karen refugees.

**Funding:** This research received no external funding.

**Institutional Review Board Statement:** Not applicable.

**Informed Consent Statement:** Not applicable.

**Data Availability Statement:** Not applicable.

**Conflicts of Interest:** The author declares no conflict of interest.

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
