# Peer review of "Protecting the Human Rights of Refugees in Camps in Thailand: The Complementary Role of International Law on Indigenous Peoples"

_laws, 1989_

Round 1
Reviewer 1 Report
This is an interesting submission that presents a thoughtful argument for the protection of indigenous Karen who are found in some nine refugee camps along the Thai-Myanmar border.
I have some points that, I believe, need to be addressed.
-- Thailand, like the majority of South East Asian States are not signatories to any of the refugee rights instruments. With the exception of the non-refoulement principle, a customary international law norm, International Refugee Law does not apply to those who are seeking asylum in Thailand. The point is made that International Human Rights Law would then be relevant for protecting the human rights of those who are in Thailand because they are signatories to the International Bill of Rights and the Convention Against Torture and other human rights instruments. It is my understanding that Thailand applies a dualist approach to treaties and that these are only binding if they are incorporated in domestic legislation. How then can IHRL apply to the Karen who are in the refugee camps along the Thai-Myanmar border?
-- At footnote 19, it is stated that the International Law of Indigenous People (ILIP) refers to ILO Convention 169, Indigenous and Tribal Peoples Convention 1989, and the UN Declaration on the Rights of Indigenous Peoples. But this Convention has only been ratified by 24 states and not by Thailand or any of the major countries such as the USA, UK, France, Russia, China, the five permanent members of the UN Security Council. The vast majority of the States of the world are not parties to the 1989 Convention.
-- The UN Declaration on the Rights of Indigenous Peoples is a non-binding instrument. And, even though Thailand voted in favour of this Declaration, Thailand is not bound by it.
-- Thailand has been ruled by military dictatorships and most recently from 2014 to 2019. Given that Myanmar is similar in this respect and has been oppressing deliberately the Rohingya and other minorities, hence, those forcibly displaced to bordering countries, it seems highly unlikely that Thailand would honour its human rights obligations to "illegal migrants." Further, Thailand has an abysmal human rights record and it is highly unlikely that they would honour their obligations to uphold the human rights of anyone, including, indigenous peoples.
For all of these reasons, I do not find your arguments persuasive.
You may wish to correct the following errors in your text.
line 56 - It should be "in order to protect."
line 163 - It is "High" and not "Higher."
line 218 - Is it not "ICESCR"?
line 652 - It should be "rather than groups."
Reviewer 2 Report
This article considers an important intersection of several areas of international law, and their application to a topical case example. The idea of exploring, in detail, how Indigenous rights and specifically the declaration, can help supplement protections available under refugee and human rights law is an interesting and worthwhile one. The topic has the potential to inform legal and policy approaches to the situation of the Karen peoples in a meaningful way. However to my mind, the article does not adequately achieve the objective it sets for itself.
The piece is also well researched, and draws upon a wide range of sources from disparate fields that span over a significant period of time to inform its discussion of the various areas of international law considered. However, much of the literature is rather dated which given the emerging body of work in these areas, and the evolving realities not least in Thailand and in international law, I would have expected to see engagement with more recent literature – both academic and grey literature incluing work of human rights committees for example.
The structure of the piece could be made more effective if the discussion of the situation of the Karen peoples was headlined, rather than left to the middle of the piece. Commencing the piece with an overview of the situation and history of, and specific issues faced by, the Karen peoples and then moving through a discussion of how each of the subject areas of international law apply in the particular context could give the piece more direction and purpose, which it currently lacks. Moreover, the article seems to try to do a number of things including prove that the Karen peoples would indeed be considered indigenous in line with existing standards; explores the role of IRL and IHRL in the context of refugees and undocumented migrants etc. I would suggest limiting this additional analysis and taking some of it to be ‘understood’ by the reader (possibly referring them to external works on this issue).
In addition, the piece seems to lack discussion of the practical consequences and application of its conclusions which results in it failing to achieve its lofty objectives. While extensive attention is devoted to considering the substance of each of international refugee law, international human rights law and international law on indigenous peoples, the actual application (or non-application) of these frameworks to the situation of the Karen peoples seems underdeveloped, and seems to have been relegated to a few sentences throughout. Moreover, the actual interaction of the legal regimes is also somewhat missing with a handful of interspersed comments here and there. Further critical analysis and discussion of the means by which international law on indigenous peoples can be used to complement international human rights law and international refugee law, how this framework should be applied to the situation of Karen peoples in refugee camps in Thailand, and what this would look like in practice, would give the article greater value.
In addition, there are a number of issues with language and phrasing which require attention. Moreover, certain key terminology is used inconsistently throughout the article, particularly the names of instruments/bodies/organizations. The article would benefit from a thorough review with specific attention to these issues. Finally, a number of basic international law concepts including binding nature of instruments (including where they are binding) are used loosely and need to be reviewed. One example that stands out is the implication throughout that the non-applicability of the refugee law framework is a weakness or failure of the refugee law regime itself.
In brief, this is an interesting topic and one that deserves attention in the form of an article. Whilst this article is well researched, it does not currently achieve the goals it has set for itself and as such is not yet in a state that is publishable in the journal. In my view, very significant re-thinking of some of the issues and re-framing of the argument is required before the article can be considered for publication.
Reviewer 3 Report
I suggest two main improvements (the second one would improve the argument in the paper):
1) provide the definition of 'refugee' under the Refugee Convention in page 1 (line 33);
2) address the issue of enforcement/or lack of enforcement of the provisions of the UNDRIP. As it has not been implemented through national legislation, how can indigenous people rely on it before a court for its violations? If enforcement is not possible, what are the advantages of integrating it with international human rights law and refugee law from a practical point of view?
Round 2
Reviewer 1 Report
I appreciate the major revisions undertaken to improve this manuscript. But I am not sure that you have addressed all my initial concerns with these revisions.
For me, there should be greater subtlety to your arguments given that it is based on such a tenuous legal basis, as you acknowledge with respect to IRL since Thailand is not a party to either 1951 Convention or its 1967 Protocol, and even though it has signed on to some of the principal international instruments in IHRL, it has not incorporated them in the laws of Thailand. Likewise, your arguments that ILIP is gaining the standing of customary international law is not persuasive, at least not to me. It may be the emerging norm by which States ought to be judged but whether it has achieved the stature of customary international law is an open question.
While I appreciate the general thrust of the argument you are presenting, which I am sympathic to, I believe the argumentation requires much more refinement to be persuasive.
There are other concerns with respect to a number of the points made throughout the manuscript such as there is no basis for collective rights in the 1951 Convention. Given that one of the grounds on which you can claim refugee protection is membership in a particular social group, then this is clearly not the case. One could likely make the same point with respect to race, religion, nationality, and political opinion, the other grounds for claiming refugee protection. While Convention refugee status may be determined on an individual basis, this does not imply it does not recognize collective rights. In addition, the UNHCR accepts refugees on a prima facie basis and it has been estimated that the vast majority of the world's refugees fall within this category.
Another example is the manner in which the term refugees is used at times seems to be conflated with "illegal migrants," which is how the Thai State considers those in the camps along its borders with Myanmar, as you point out. As you note, neither Myanmar nor Thailand recognize anyone as refugees because they do not wish to be parties to the 1951 Convention or 1967 Protocol. Then is there any legal recognition of those that are in the refugee camps along the Thai-Myanmar border as "refugees"? Would it not be better to argue that all those who are forcibly displaced and in this encampment situation are better protected under IHRL?
A further consideration here is that all member States of the UN are obligated to cooperate with the UNHCR in the fulfilment of its responsibilities to protect refugees, irrespective of whether they are parties to any refugee rights instruments. Presumably, Thailand is prepared to cooperate with the UNHCR in this regard. And, if so, can they persuade the Thai government to do more for the Karen who find themselves living in the Thai camps? (See https://www.unhcr.org/th/en/unhcr-in-thailand.)
